# Will Peri-Urban *Cydia pomonella* (Lepidoptera: Tortricidae) Challenge Local Eradication?

**DOI:** 10.3390/insects11040207

**Published:** 2020-03-27

**Authors:** Rachael Horner, Georgia Paterson, James T.S. Walker, George L.W. Perry, Rodelyn Jaksons, David Maxwell Suckling

**Affiliations:** 1The New Zealand Institute for Plant and Food Research Limited, Private Bag 4704, Christchurch Mail Centre, Christchurch 8140, New Zealand; rodelyn.jaksons@plantandfood.co.nz (R.J.); max.suckling@plantandfood.co.nz (D.M.S.); 2The New Zealand Institute for Plant and Food Research Limited, Private Bag 1401, Havelock North 4172, New Zealand; gpat568@aucklanduni.ac.nz (G.P.);; 3School of Environment, The University of Auckland, Private Bag 92019, Auckland 1142, New Zealand; george.perry@auckland.ac.nz; 4School of Biological Sciences, The University of Auckland, Private Bag 92019, Auckland 1142, New Zealand

**Keywords:** eradication, suppression, public, biosecurity, *Cydia pomonella*, sterile insect technique, peri-urban, pheromone trap, market access, apple, *Malus domestica*

## Abstract

Codling moth, *Cydia pomonella* (Lepidoptera: Tortricidae), is a phytosanitary pest of New Zealand’s export apples. The sterile insect technique supplements other controls in an eradication attempt at an isolated group of orchards in Hawke’s Bay, New Zealand. There has been no attempt in New Zealand to characterize potential sources of uncontrolled peri-urban populations, which we predicted to be larger than in managed orchards. We installed 200 pheromone traps across Hastings city, which averaged 0.32 moths/trap/week. We also mapped host trees around the pilot eradication orchards and installed 28 traps in rural Ongaonga, which averaged 0.59 moths/trap/week. In Hastings, traps in host trees caught significantly more males than traps in non-host trees, and spatial interpolation showed evidence of spatial clustering. Traps in orchards operating the most stringent codling moth management averaged half the catch rate of Hastings peri-urban traps. Orchards with less rigorous moth control had a 5-fold higher trap catch rate. We conclude that peri-urban populations are significant and ubiquitous, and that special measures to reduce pest prevalence are needed to achieve area-wide suppression and reduce the risk of immigration into export orchards. Because the location of all host trees in Hastings is not known, it could be more cost-effectively assumed that hosts are ubiquitous across the city and the area treated accordingly.

## 1. Introduction

Codling moth, *Cydia pomonella* L., is a pernicious, almost cosmopolitan pest of apples (*Malus domestica*), with one or more generations per year, depending on the climate [1]. It is believed to be native to Central Asia and has spread with the cultivation of apples to most temperate regions of the world, including Europe, China, North and South America, South Africa, Australia, and New Zealand [2]. Intensive management practices, including insecticide applications, are generally required for codling moth control worldwide, but control is increasingly being achieved by the use of sex pheromone traps for population monitoring, as well as pheromone dispensers for mating disruption [3]. Increasingly, the term area-wide is being used to represent a higher degree of coordination than the orchard level [4]. The success of an area-wide integrated pest management program relies upon a holistic and multidisciplinary approach to achieve effective management. The crucial part of an area-wide strategy is an emphasis on preventing the existence of foci of infestation from which recruits can re-establish damaging densities of the pest population [5]. In British Columbia, Canada, a pioneering sterile insect technique (SIT) program was developed based on the extensive field and laboratory research of Proverbs [6,7,8,9]. Other early researchers also showed the potential for SIT against this species [10,11]. The SIT involves the sterilization and release of mass-reared insects into the environment, enabling sterile males to compete with wild males to mate with wild females, reducing the number of offspring [12].

The current Canadian SIT program was first established to mass-rear and release sterile codling moth into local apple orchards, as a replacement for organophosphate insecticides. The initial stated goal was eradication. However, eradication was not achieved in British Columbia, in part because of immigration from peri-urban host trees [13]. Unmanaged orchards and backyard host trees are now known to have an impact upon an area-wide integrated pest management or AW-IPM codling moth program, with immigration occurring into orchards already under active suppression or eradication [14,15]. Spatial analyses of pheromone trap catches of codling moth in heterogeneous agro-ecosystems in Italy have also shown that trap capture foci were clumped in productive orchards and in zones containing host trees [16]. However, it is possible to eradicate codling moth. It has been successfully eradicated (prevented from establishment despite incursions) in Western Australia [4], and an eradication program has been pursued in peri-urban Brazil through host tree removal and other tactics [5]. These successes and failures provide immense value to the current era in IPM.

The export-dominated New Zealand apple sector has responded to increasing market demands for both ultra-low residue fruit and strict phytosanitary requirements from more than 60 countries and has successfully dealt with codling moth for over 100 years [17]. Implementation of the Integrated Fruit Production (IFP) program in New Zealand apple crops, followed by ultra-low residue initiatives [18] has led to the elimination of organophosphate, neonicotinyl, and carbamate insecticide use, and resulted in the widespread adoption of selective pest management [17]. This program has substantially reduced the status of previously important pests [17]. However, codling moth remains a major pest of quarantine concern on apple exports to certain high-value eastern Asian markets, where it is absent. Some markets require mandatory methyl bromide fumigation and there is zero tolerance for live codling moth in fruit. The presence of a single codling moth in an export shipment can severely impact market access for all New Zealand apple exports. To maintain access to these markets, management of export orchards is based on a regulatory program. It is a ‘systems approach’ to codling moth management and results in ultra-low levels of codling moth on orchards. Non-export orchards generally have fewer codling moth management inputs and therefore higher codling moth trap catches, relative to export orchards.

The Hawke’s Bay region is the most important apple growing district in New Zealand, and has a relatively warm and dry climate as a result of it being sheltered by mountain ranges [19]. Here, one extended generation of codling moth is common, with emergence from October to February and peak emergence in mid-December [20]. New Zealand researchers are seeking more sustainable methods for codling moth suppression for resistance management, and because on-orchard control is becoming increasingly important for market access as post-harvest controls, such as methyl bromide, are removed from use [21]. The requirement for strict, specified orchard management practices to keep codling moth populations to negligible levels in orchards has resulted in the trialing of sterile codling moth releases in New Zealand for the first time [22]. A pilot codling moth eradication program has been operating in a cluster of 390 ha of apple orchards since 2014. The cluster is isolated from other orchards, in Ongaonga, Central Hawke’s Bay (CHB), New Zealand (Figure 1). A combination of tactics is being used to reduce populations of codling moth in orchards, including mating disruption, insecticides, and the sterile insect technique. 

Six large corporate orchards began this program after a history of low codling moth activity after more than 15 years of adhering to the regulatory export program (on average less than 5 moths per 10 traps in a whole season), and a seventh large organic orchard had comparatively higher codling moth activity. Results are promising, with large drops in codling moth populations in these orchards, but an eradication attempt necessitates consideration of any neighboring populations of codling moth [14,15]. Large and consistent reductions in the CHB orchard population have led to questions about the potential for expanding the use of the SIT to the main apple production areas of the Heretaunga Plains as part of an area-wide program, but the entire population of codling moth needs to be included in an area-wide program [13]. 

While there are very few unmanaged orchards near commercial orchards in New Zealand, due to strict management, towns are frequently characterized by houses with gardens, where apple, pear, and walnut trees are frequently planted by homeowners. However, many homeowners’ ability to manage codling moth is probably limited. We set out to improve the understanding of the size and extent of the peri-urban population in host trees around the pilot eradication area of Central Hawke’s Bay, including a settlement within the SIT program region (“Ongaonga”, pop. 200) [23], to understand the potential impact that moth movement across the largely pastoral landscape could have on the eradication program. Expansion of the program to the wider Hawke’s Bay area could be considered if this pilot eradication was successful and the economics, logistics, and likelihood of success of a wider SIT program were favorable. Therefore, the peri-urban codling moth population in the wider Heretaunga Plains area were also investigated, centered on the city of Hastings (population 73,245) [23], where we had also investigated homeowner attitudes to the sterile insect technique [24].

We hypothesized that codling moth populations would be ubiquitous where traps were set in CHB and peri-urban Hastings, and that like elsewhere, higher catch rates would be observed in peri-urban areas than in nearby apple orchards, where populations are generally under active suppression. Finally, we expected that codling moth trap catch rates would be correlated with the presence of host trees in the vicinity. We set sex pheromone traps in peri-urban home gardens and compared trap catch rates with those in export and non-export orchards where pre-set thresholds for trap catches are used by the apple industry to determine intervention, and obligatory trap records were supplied by fruit growers [17]. 

## 2. Methods

### 2.1. Mapping Host Trees

For a 9-km radius around the orchards involved in the pilot eradication program in Central Hawkes Bay, a thorough survey of apple (*Malus domestica* L.) and pear (*Pyrus* L.) trees was conducted at bloom time. Common walnut trees (*Juglans regia* L.) are a large, well-established ornamental tree species in New Zealand, and a known host of codling moth, and locations of these trees were also recorded in the survey. In Hastings, the locations of apple, pear, and walnut trees were identified where possible, by talking to landowners when traps were placed [24], and any known host trees on neighboring properties were noted. Walnuts were extensively mapped in 2011, but data were not published. This dataset (*n* = 336 walnut trees), as well as further data collected about backyard host trees during the survey, were used to investigate the effect of host location on trap catch in Hastings. Other known host trees, such as quince and crabapple, are rare in New Zealand and were not observed in the survey, and thus references to host trees refer to apple, pear, and walnut trees. 

### 2.2. Traps

Red delta traps (28 by 20 cm) supplied by Desire™, Etec Crop Solutions Ltd. (Auckland, New Zealand) made of Corflute™ plastic [25] were baited with 1 mg codlemone rubber septa lures (Desire™, Etec Crop Solutions Ltd., Auckland, New Zealand), which is the New Zealand apple industry standard, except where mating disruption is applied to orchards, and rubber septa were loaded at 10 mg of codlemone. Pheromone lures were changed every 6 weeks. All traps were set at approximately 1.5 m in height and locations were recorded using a GPS device. Trap catches were standardized to moths/trap/week.

### 2.3. Trapping Locations

Two areas, with high and low densities of human habitation, were studied. A grid with a total of 200 traps over ~1500 ha (300 m spacing or one trap/9 ha) was operated throughout Hastings city except in the central business district (lat. −39.639, long. 176.846). The peri-urban area is characterized by extensive boundaries between residential and apple production areas, with all residences in Hastings being within 2.5 km of an orchard. Traps were installed with the permission of homeowners, who were also asked questions as part of a survey of peri-urban community attitudes to the pest and the SIT [24]. Where a host tree was available, the trap was placed in that tree (*n* = 11 for apple, *n* = 1 for pear, and *n* = 3 for walnut). If there was no host tree, another non-host tree was selected for trap placement. The non-host trees were of a variety of species. The traps were checked, and sticky bases changed fortnightly from 13 November 2017 to 23 February 2018 (15 weeks). 

We also operated 28 traps in apple and walnut host trees (*n* = 19 for apple and *n* = 9 for walnut) identified within 9 km of the target eradication orchards in the rural area of Ongaonga (population c. 200) (lat. −39.916148, long. 176.419543) from 26 October 2018 to 13 March 2019 (20 weeks), with traps checked and bases changed fortnightly. The seven orchards are each surrounded by pastoral farms. Trap data from these orchards were not used as the use of insecticides, mating disruption, and sterile insects has reduced moth catches to ultra-low levels (one orchard had only 1 moth per 100 ha per year). 

### 2.4. Orchard Trap Catch Data from the Hawke’s Bay Growing District

Pheromone trap catch data for 4292 traps located in export orchards and 286 traps located in non-export orchards in Hawke’s Bay were extracted from a grower database. Export orchards have one pheromone trap per hectare. About 50% of export orchards are under mating disruption and therefore operate lures with 10-fold more codlemone than standard lures. Non-export orchards have one pheromone trap per 2 hectares, and use an insecticide-based program. These traps were operated from late October to late March. 

### 2.5. Statistical Analysis

All statistical analysis was carried out using R version 3.5.3 [26]. A Mann–Whitney U test was conducted to detect any evidence of a difference between traps in host trees (*n* = 15) and traps in non-host trees (*n* = 185). This test was undertaken because the distributions of the trap captures of the host and non-hosts were non-normal, thus a non-parametric test was required. To study the spatial distribution of the total trap captures, an exponential variogram model was fitted to the data using the geoR [27] package in R. A variogram model examines the correlation between points, with the level of correlation depending on distance alone. Variogram models state that points become more dissimilar with increasing distance. Using the variogram model, it is possible to conduct spatial prediction into unobserved locations (spatial interpolation) via kriging [28]. Additionally, probabilities of exceeding a certain threshold can also be calculated. In this study, we calculated the probability of a location exceeding five captures to allow us to identify areas of high codling moth population.

## 3. Results

### 3.1. Hastings Densities and Counts

In total, 954 male codling moths were caught across the 200 traps in peri-urban Hastings (Figure 2), or on average 0.32 moths/trap/week. Of the 200 traps, only 12 traps (6%) caught more than one moth per week, while 32 traps (16%) caught zero codling moths across the trapping period (Figure 3). In export orchards, pheromone traps caught an average of 0.17 moths/trap/week, which is half the trap catch of traps in the peri-urban area. In non-export orchards, pheromone traps caught an average of 1.62 moths/trap/week, which is 5-fold higher than in the peri-urban traps. 

There was a significant difference in the trap catch rate between traps placed in host trees (*Median* = 0.69 moth/trap/week) and those placed in non-host trees (*Median* = 0.19 moths/trap/week) (W = 2323, *p* = 4.31 × 10^−5^) (Figure 4). It was calculated that 96% of traps had a known host tree within 1 km and 64% within 500 m. The average distance to the nearest host tree was 434 ± 289 m (mean ± 1 SD). All traps placed in host trees caught some codling moths. 

The spatial interpolations of the variogram models are given in Figure 5 and Figure 6 and show evidence of spatial clustering in the Hastings data. Figure 5 gives the probability of a location exceeding a total trap capture of five. The darker areas are of regions of high probabilities, and thus can be considered as areal hotspots. Figure 6 shows the estimated trap catch of a location, with the darker regions showing higher estimated catches. The majority of hotspots occur in close proximity to host trees. However, in some cases, the predicted hotspots are not in the vicinity of known host trees. 

### 3.2. Central Hawke’s Bay Densities and Counts

In total, 331 codling moths were caught across the 28 traps (0.59 moths/trap/week) deployed in host trees identified in the region of the apple orchards in Central Hawke’s Bay, where eradication is being attempted. Twenty percent of traps in host trees did not catch any moths, suggesting that some host trees in CHB apparently do not have resident populations of moths. In contrast, all traps in host trees in Hastings caught moths (Figure 7 and Figure 8). The trapping densities in CHB and were different to that of Hastings, as the traps were in a grid in Hastings, whereas in CHB traps were confined to host trees, which had a non-uniform distribution.

## 4. Discussion

Peri-urban populations of apple and walnut trees pose a threat to production orchards through the movement of codling moths, and unmanaged trees can affect nearby managed orchards [5]. In a study conducted in Washington, USA, Butt et al. (1973) [11] concluded that most of the moths on commercial orchards were sourced from areas of untreated apple trees that were in close proximity. Prior to our study, no information about peri-urban codling moth populations was available for New Zealand, and the pioneering Canadian program had provided clear signals of their importance. The ambitious plan to attempt a pilot eradication as a research project across multiple orchards (total area of c. 390 ha) necessitates the consideration of any neighboring resident populations of codling moth. Our study shows that a codling moth eradication plan for Central Hawkes Bay is ambitious but may be possible due to the grassland landscape and the small number of alternative sources of codling moth observed in the vicinity of the orchards. Moths would have to travel large distances between host trees. Curiously, 20% of traps in host trees in this rural area did not catch any codling moths. This could have occurred due to the presence of un-infested young trees and a low and patchy population density. However, our study shows overall that there are no areas of Hastings, in the main apple growing district, from which codling moth is absent. Complete area-wide treatment would be necessary to create an area of low pest prevalence, including all population centers where similar densities to Hastings of residential apple and walnut trees would be expected to harbor the pest. Pest populations of codling moth outside orchards will continue to pose a risk to the viability of fruit in managed orchards if not effectively managed. Re-invasion will also remain a risk from unmanaged peri-urban populations, even if negligible codling moth risk is achieved in orchards in any given year.

In Hastings, catch counts were significantly higher in host trees than in non-host trees and all traps placed in host trees caught moths. Similar observations have been made in traps placed in host trees versus non-host trees for gum leaf skeletoniser, *Uraba lugens* [29], and oak processionary moth (*Thaumetopoea processionea*) [30]. Moths can seek out high-quality hosts, and respond to volatile cues with a highly refined host location system based on olfactory receptor neurons. Male moths integrate host odor and sex pheromone cues to find mates and food, which, in turn, raises sex pheromone trap catch rates in host trees [31,32,33]. This behavior demonstrates the benefit of placing traps for codling moth in host trees and also presents the possibility of using kairomones for codling moth [34,35] in new ways, as part of the suppression program. 

It is not known exactly how many backyard apple and walnut trees there are across peri-urban Hastings, and this is a weakness of the study. However, we attempted to locate hosts during the deployment of the traps. There are approximately 9975 occupied dwellings located in the selected study areas in Hastings [23]. Seven of the 86 (c. 8%) dwellings surveyed had a host tree on their property [24], so it would be expected that there would be a much larger number of host trees throughout the Hastings region (potentially up to 800 host trees based on a simple linear upscaling). The identification of all host trees would facilitate management of codling moth in the Hawke’s Bay region but would be costly. Our spatial interpolation of the variogram models showed that while the majority of hotspots are shown in close proximity to known host trees, there are some hotspots that are not in the vicinity of any known host trees. These areas are likely to have one or more host trees present and allow us to better target further host tree identification. Because codling moth were widespread across Hastings, it could be more cost-effectively assumed that hosts and the pest are ubiquitous across Hastings and the area treated accordingly.

Butt et al (1973) [11] inferred that the number of codling moths that overwintered was significantly reduced when neighboring uncontrolled orchards and backyard apple trees were either removed or the pest controlled. In Brazil, the removal of host trees in peri-urban areas and replacement with non-host trees led to a significant decrease in codling moth populations, as evidenced by the decrease in the number of catches in pheromone-baited traps used for monitoring, and eradication in an urban area was realized [36]. The Okanagan-Kootenay Sterile Insect Release program carried out the removal of several thousand peri-urban host-trees, along with fruit stripping and incentivized tree replacement. While they significantly reduced the numbers of codling moth, eradication was not achieved, and peri-urban populations were recognized as a contributing factor to that [15].

The removal of backyard host trees would be ideal for codling moth control and eradication. However, in our related study of community attitudes to suppression tactics [24], most homeowners did not support this option. Instead, fruit could be stripped for a limited period and the location of host trees could be used as foci for management techniques, such as the aerial area-wide release of sterile codling moths. A sustained combination of area-wide tactics in orchard and peri-urban areas, such as tree banding, mating disruption, insecticides, and sterile insect releases, would facilitate suppression. 

Every private dwelling in peri-urban Hastings is within 2.5 km of an apple orchard. Codling moth adults can fly several kilometers, depending on wind conditions and topography [37,38], so the on-orchard control of codling moth would never eradicate the wider regional population. The tendency for movement of adult codling moth in a peri-urban environment may be limited by landscape features, such as host plant availability, as well as surface roughness from houses and property fences. There may only be a minimal number of dispersing individuals that connect distant populations with locally high populations. However, these rare long-distance dispersers are extremely important for pest species spread and maintaining genetic homogeneity. During a suppression program, movement corridors could be considered as foci for treatment and strips of non-host trees around orchards could be exploited as barriers to reinvasion [39]. 

The pheromone traps only catch males, so the results discussed here are valid only for adult males. Management of codling moth females may be possible once relevant observations become available, perhaps with the use of female attractants [34,40,41]. Comparisons between codling moth populations in the three habitat zones of peri-urban areas, export, and non-export orchards are complex and should be made with caution, because of the differences in trapping density. In the peri-urban area, traps were set at one trap per 9 hectares, while in export orchards, traps were set at one trap per hectare, and in non-export orchards, the density was two traps per hectare. The peri-urban pheromone traps had twice as high a catch rate as those in export orchards. While the behaviorally effective plume reach is thought to be <5 m for codling moth, the total trapping area for a single trap is estimated at approximately 21 ha in contiguous apple orchards [42]. Trap competition would have been much higher in export orchards than in peri-urban areas, as the traps were much closer. Codling moth in half of the export orchards are also all under competitive attraction from a high density of dispensers of synthetic pheromone; thus, the frequency that male moths find female moths or pheromone traps is reduced, as they first attract and then de-activate males for the remainder of the night [42]. To compensate for the disruption effect, pheromone traps were loaded with 10-fold more codlemone, so any direct comparison between non-export and peri-urban trap catches must be made cautiously. In non-export orchards, pheromone traps caught an average of 1.62 moths/trap/week, which is 5 times higher than in the peri-urban traps. These orchards are generally not under mating disruption, with fewer insecticide applications throughout the season. The higher number of host trees under lower levels of control in these orchards accounts for the higher trap catches compared with the peri-urban area and particularly the export orchards. 

While the application of the sterile insect technique against codling moth has yet to be undertaken in peri-urban or urban areas, there have been previous moth trapping and eradication programs in urban New Zealand. These programs targeted several invasive species [43], with one other sterile insect program where the population was eradicated by a combination of tactics, entirely within the city [44]. A recognition of the concerns of the public is an essential part of operating pest suppression in the peri-urban space [24].

## 5. Conclusions

We mapped host trees around the pilot eradication orchards in CHB and found large, uncontrolled populations of codling moth. However eradication could be possible due to the small number of host trees and the large distances between host trees in the mostly grassland habitat surrounding the orchards. Due to the large and ubiquitous codling moth population found across Hastings, eradication would be difficult. Because the location of all host trees is not known, it could be more cost-effectively assumed that hosts are ubiquitous across Hastings and the area treated accordingly. External populations of codling moths will continue to pose risks to the export of fruit from orchards, if not effectively managed. Re-invasion would remain a risk from unmanaged peri-urban populations, even if suppression or even local eradication is achieved in orchards in a given year.

## Figures and Tables

**Figure 1 insects-11-00207-f001:**
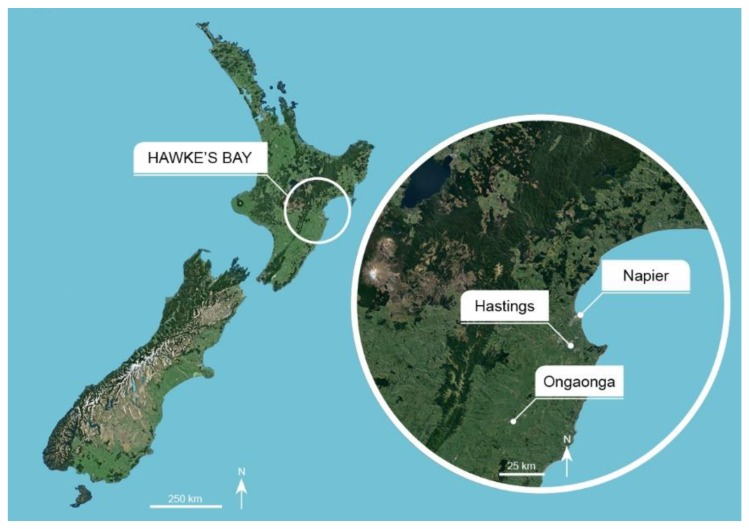
The location of Hawke’s Bay in New Zealand and the two study sites of Hastings on the Heretaunga Plains and Ongaonga in Central Hawke’s Bay. Map image courtesy Google Earth (accessed on 4 June 2019).

**Figure 2 insects-11-00207-f002:**
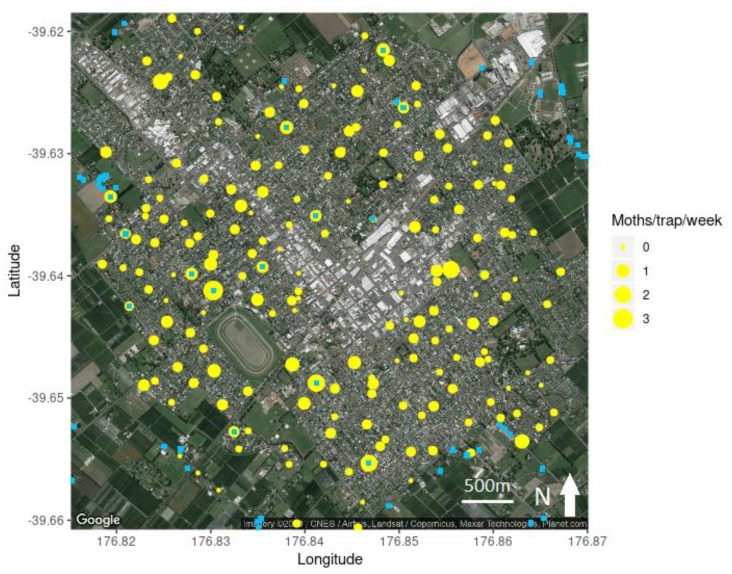
Distribution of the male codling moth (*Cydia pomonella*) catch rate across peri-urban Hastings alongside the location of known host trees (blue squares *n* = 351) and the location of the sex pheromone traps (yellow circles; *n* = 200). Map image courtesy Google Earth (4 June 2019).

**Figure 3 insects-11-00207-f003:**
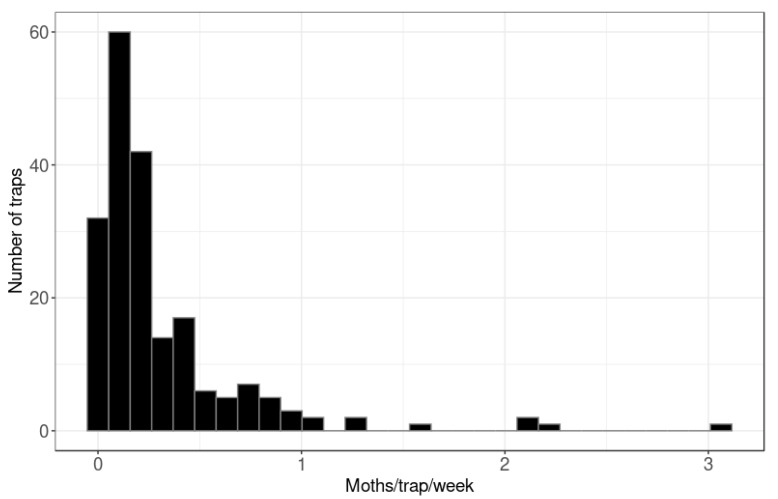
Distribution of the number of codling moth (*Cydia pomonella*) caught in pheromone traps per trap per week in peri-urban Hastings (*n* = 200).

**Figure 4 insects-11-00207-f004:**
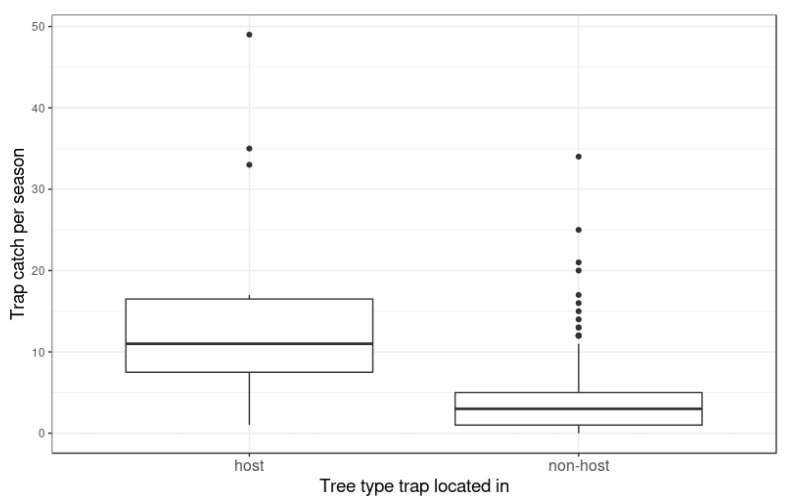
Number of codling moths (*Cydia pomonella*) caught per trap per week when traps were positioned in host (*n* = 15) and non-host trees (*n* = 185). The ends of the boxes are the upper and lower quartiles and the bold line shows the median. The whiskers extend up to 1.5 times the interquartile range from the top of the box to the furthest datum within that distance. If there are any data beyond that distance, they are represented individually as points (‘outliers’).

**Figure 5 insects-11-00207-f005:**
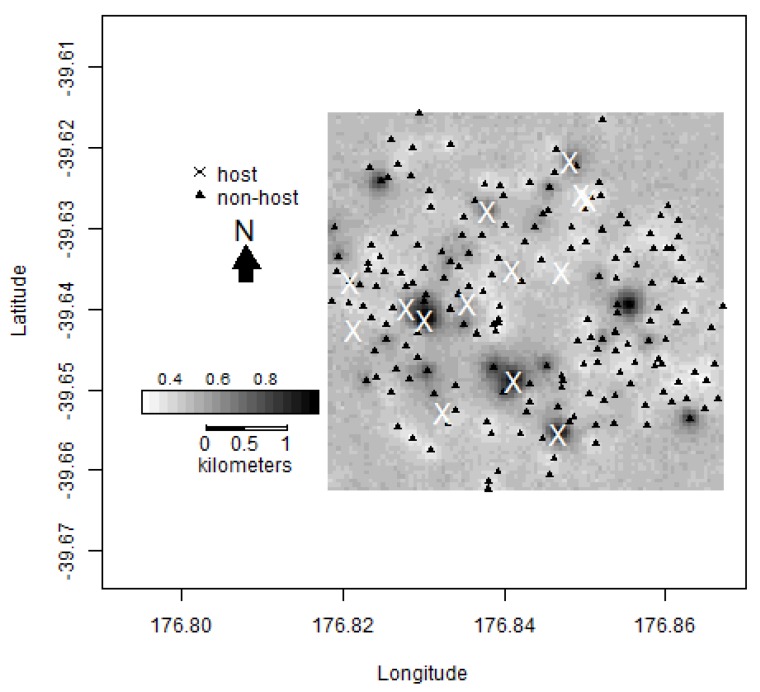
Spatial interpolation giving the probability of a location exceeding a total seasonal trap catch of five male codling moths (*Cydia pomonella*). The darker areas are of regions of high probabilities within Hastings, and thus can be considered as areal hotspots. The crosses signify the locations of traps in host trees (*n* = 15) and the triangles signify the locations of traps in non-host trees (*n* = 185).

**Figure 6 insects-11-00207-f006:**
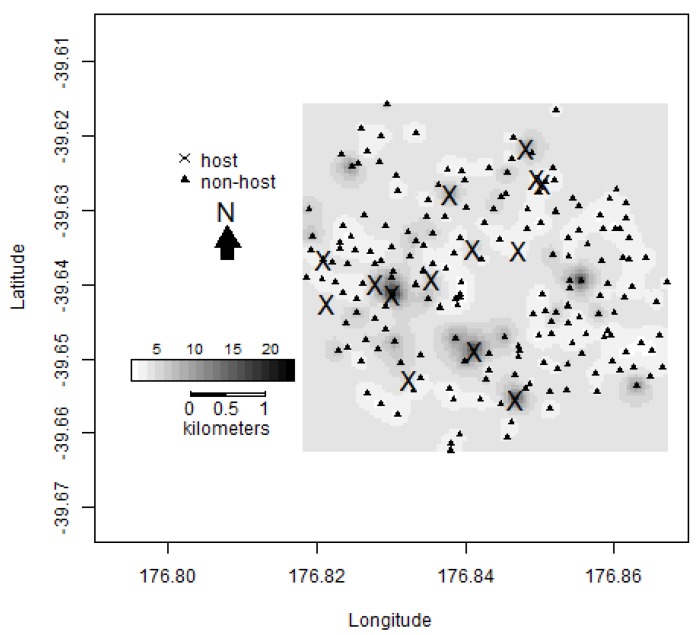
Spatial interpolation of male codling moth (*Cydia pomonella*) trap catches showing the estimated trap catch of a location, with the darker regions showing higher estimated catches within Hastings. The crosses signify the locations of traps in host trees (*n* = 15) and the triangles signify the locations of traps in non-host trees (*n* = 185).

**Figure 7 insects-11-00207-f007:**
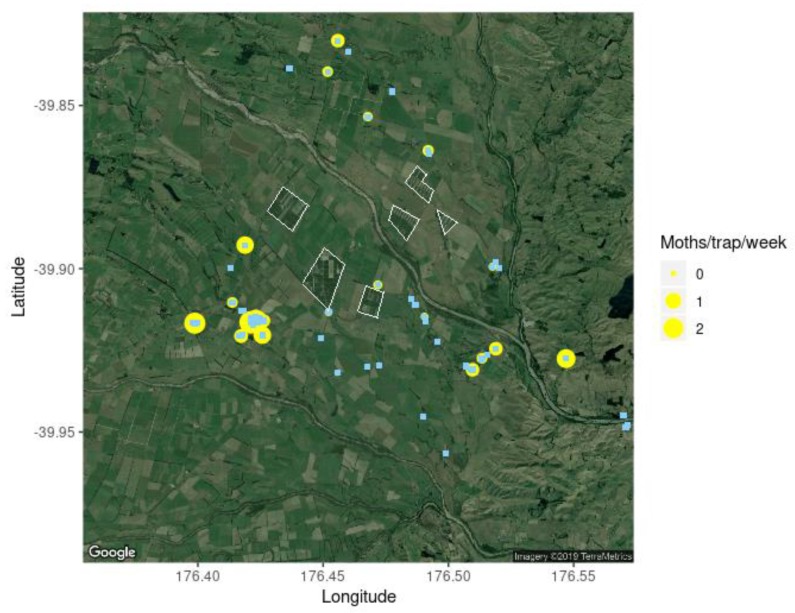
Distribution of the male codling moth (*Cydia pomonella*) catch rate across peri-urban Ongaonga alongside the location of known host trees (blue squares *n* = 62) and the location of the sex pheromone traps (yellow circles; *n* = 28). The locations of the target eradication orchards are indicated by white polygons. Map image courtesy Google Earth (4 June 2019).

**Figure 8 insects-11-00207-f008:**
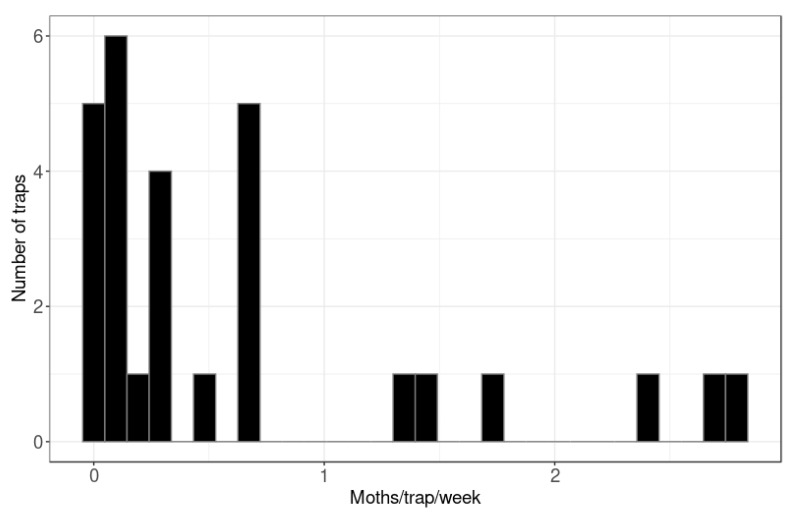
Distribution of the number of male codling moths (*Cydia pomonella*) caught in pheromone traps (*n* = 28) in peri-urban Ongaonga.

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
