# Peer review of "Will Peri-Urban Cydia pomonella (Lepidoptera: Tortricidae) Challenge Local Eradication?"

_insects, 2020, doi:10.3390/insects11040207_

Round 1

Reviewer 1 Report

The article arises from a study conducted during one season as part of a possible eradication project for codling moth in NZ. The aim was to assess challenges arising from the presence of host trees of codling moth. A brief survey was conducted, and pheromone traps were placed in some locations. The results were summarised briefly, and a form of spatial analysis conducted.

In view of the recommendation to reject, rare for this reviewer, possible improvements are given below for consideration by the senior author(s) in future work.

For an international journal, the authors need to provide a short description of the life-cycle of codling moth in this region of NZ, and of any anomalies in the climate or landscape.

The article is surprising in its naïveté with regard to the importance of extra-orchard host trees. The authors seem surprised as to their overall importance in challenging an area-wide program. In planning, they missed the extensive literature on problems for area-wide programs and SIT arising from host trees of codling moth. First described extensively in the 1960s and 1970s from North America (articles by Proverbs, Hutt, White, et alia) and Europe (summarised by Dickler in 1978 Mitt. Biol. Bundes. Land und Forst. vol 180 pp. 120), and reinforced when eradication was abandoned in British Columbia for this reason, in 2000 (reviewed recently in Insects). Such articles describe the importance of understanding exactly where all host trees are located, and ensuring their correct treatment.

The methods employed in this study are superficial and, as described, are not fit to purpose. No real ground inspection appears to have been conducted at bloom time (when readily apparent), even in a small portion of the area, and non-host trees are repeatedly mentioned but defined nowhere. Banding of trees and fruit inspection would have been more correct in identifying host plants than inferred populations from traps, which moths may have arrived from other sources than the trap tree.

The Introduction should have a brief summary of the appropriate international literature for the questions being considered. In particular, the challenges found in area-wide management and what the specific objectives and hypotheses of the present work set out to test. Instead, two underwhelming goals, almost unworthy of publication, are given in L 93-96.

The conclusions are only half drawn from the study. Of the first three sentences, only the second pertains to the study and others are discussion points seemingly driven by a non-scientific agenda.

The article may have been of much more value if they had compared the approach taken in this project to the reported success of SIT against the Painted Apple moths. Instead, nothing new is learnt of the well-known codling moth peri-urban problem.

METHOD Given the knowledge of the problem published from other regions, the study was not well designed or described. The analyses are problematic. The conclusions bear little resemblance to the results. The methods, plant species, traps, lures, and statistical analysis are not described in sufficient detail. Example - methods are not sufficiently described, or require further explanation or reanalysis. Three examples of problems in wording follow:

1 Delta traps are not described with dimensions or suppliers, merely a reference to a regional publication citing a comparison; lures are not described with respect to delivery system for 1mg and 10mg codlemone, so reader has no idea as to whether or not a 6 week change period is appropriate (rubber septa vs. plastic rope, and so on).

2 Two classes of trees, host and non-host are mentioned repeatedly in the text, but not described beyond “such as …” three spp. listed for host trees. The “non-host tree” spp. are not described. Together, the former renders much of the text as unclear.

3 A bewildering mix of trap and orchard terminology is varied without explanation: traps in export orchards, traps in non-export orchards, non-export targeted orchards (do these differ from others?), commercial orchards, commercial export orchards, commercial non-export orchards, peri-urban traps, urban traps, host tree traps, non-host tree traps, export and non-export traps, are all mentioned. Define each at first use and stick to a small clear list of terms.

One analytical example: spatial analysis using variograms and kriging requires special handling for biological systems. The theory is derived from analysis of static geological formations, and so biologists are required to analyse such data according to a given period of activity such as flight periods (often weekly). If more than one generation occurs within a study, then per generation at the maximum. Such occurred in various articles by the group of Trematerra, for example, one of which is cited herein. In the present study, the authors appear to have conflated all results into a single analysis per region. If they had earlier analysed the results periodically or by generation, and stated that there were no significant differences when compressing into one variogram per region, okay. However, they have not, and it is possible that they do not understand spatial analysis and may have missed important results.

The authors make some key methodological errors which may have been avoided, such as not performing a painstaking ground-truthed survey of host trees. Repeated experience has shown the importance of finding every tree, up to 1.5km from protected orchards. Also, they refer to numbers of moths per trap. The infestation rate of host trees is likely of more value in the long run because population levels are a result of variable and unknown levels of control in urban areas, whereas the moth can increase 50-100 times per generation (i.e. very rapidly).

The conclusions are already well-known to the limited number of people interested in eradication attempts for well-established populations of codling moth or other Lepidoptera.

The authors are encouraged to rewrite some sections for clarification. Break up run-on, lengthy, sentences into short phrases. Notably L 131-134, L 154-157, L 92-195

The analysis appears to be incomplete and the figures should be revised. Kindly change the colour choice for colour-blind folks, and change the purple crosses which are impossible to see (this reviewer is not colour-blind) in Figs 2 and 7 . Redraw Figs 3, 4, 8, with more attention to borders, letter size, and scale units, for clarity. Revise Figs 3 and 8 onto one graph using different hatching for each, and add more units along the x axis for clarity. Putting them onto one chart would permit better comparison even if two y scales are used (on each side), and save space.

Specific comments

L 40 For scientific balance, bring forward from L 67, and 247 251, the failure to eradicate in British Culumbia, owing in large part to underestimation of number of host trees outside orchards and the effort required in peri-urban management.

L 54 55 Confusing -  methomyl bromide is presumably employed for post-harvest disinfestation (?) whereas this article deals with orchard and urban management.

L 57 What are negligible levels in NZ may be high somewhere else or vice versa – give some values.

L 69 As mentioned under general comments, the authors are advised to reread the articles on Cydia spp. by Trematerra and to reanalyse their own data spatially by each generation present, at minimum. A recent review of codling moth SIT in Insects (Thislewood and Jodd) emphasised the importance of spatial interactions and of host tree management.

L 73 74 Suggest numeric values of codling moth be added (see comment L 57).

L 85 and elsewhere. The authors employ in the Ms. several vague compund terms of eradication, such as local, pilot, or orchard. These should be defined once at first use and stuck to thereafter.

L 105 Unclear that the lures work for 6 weeks from the absence of description in methods.

L 111 Define “extensive boundaries”

L 115 Give table of host tree species or describe in text

L 124 126 Sentence including “ along with the remaining ….” lacks meaning

L 129 How were non-host trees mapped and used? Never described.

L 154 157 Which orchards are referred to – export, non-export, etc.?

L 161  Report number of purple crosses, as given for traps. Note, the term n is usually small and italic if representing number

L 163 Are these urban, peri-urban, or urban only (see terminology comment above)?

L 166 Absence of definition of non-host trees results in a meaningless comparison.

L 171 onwards and Fig 4. What are host and non-host trees?

Figures 5, 6 Please describe in text or Methods, what the host and non-host images portray and why you show Hastings, then Urban Hastings, in the Figures. Do these differ?

L 185 areal?

L 192 Discuss why 20% of traps in host trees do not catch. This is a potential problem for the study – was it a trap/lure problem, arising from homeowner pesticide use, or juvenile trees.

L 194 195 The importance of grassland and trapping densities in affecting trap catch in host trees is unclear.

L 210 212 Move to the Introduction

L 219 Meaningless until explained what host and non-host trees are

L 220 227 Rather than describe lures, suggest that the authors describe prior results from the literature of area-wide management or local/orchard/ eradication of authors cited earlier by this reviewer.

L 228 231, 234 236 Good. This was the critical part of the study, according to prior researchers and experience in Canada and elsewhere. Commendable that the authors acknowledge a problem in their approach.

L 240 241 This comment is the single novel element in the entire work, critical for the conclusions and the implications are enormous. Presently acknowledged only in the second sentence of the conclusions.

L 251 There are many poor choices of citations in the article, and this is an example. Works by Proverbs, and a recently review in Insects, are far more appropriate than a cactus moth book chapter.

L 252 257 Failure of the BC SIT Program was attributed in large part to underestimating numbers of host trees, and failure to achieve necessary SIT ratios owing in part to extra-orchard sources of moths.

L 257 261 Scientifically accurate to use terms “possible” or “may” here, because the notion of F1 sterility is merely theoretical and unproven for codling moth. In addition, if it did occur, theory predicts the presence of increased trap counts, and of increased fruit damage. Finally, the recent Canadian program abandoned urban releases owing to lack of efficacy in spot infestations, as discussed in a recent review. Also identified as problem in biological works of Geier in Australia, earlier.

L 278 The values shown are estimated from work in large contiguous orchards and unrelated to an urban situation.

Reviewer 2 Report

From my point of view the manuscript provide very useful information, so it deserves to be published after few minor revision of the text.

These editing errors are listed in the attached file.

Author Response

Dear reviewer,

Thankyou for your review. Please find below the minor changes made to the manuscript.

Regards,

Rachael Horner

L164. Just one final point.

Amended

L185. “Areal” should be “a real”

Not amended. Definition of areal: of, relating to, or involving an area

Line 269. It should be “During a suppression”

Amended

Line 319 onwards. All the references has their number in duplicate.

Amended

Reviewer 3 Report

The conducted research is very current and meets the set goals. It is obvious that the authors are managing the issue, especially the possibility of reducing the population of Cydia pomonella in the non-agricultural area.

Author Response

Dear reviewer,

We appreciate your review. There were no changes required.

Regards,

Rachael Horner

Round 2

Reviewer 1 Report

VERSION 2

The authors substantially revised the ms. to one worthy of review.

As provided, I see no line numbers with which to compare the versions or refer to the authors revision and comments. Consequently, mine will be more general.

Abstract

“..suggesting that the catch distribution of C. pomonella was not random.” Why would it be random, based on all prior work with CM for the last 100 years? Suggest replacing this comment with something useful from the study results.

“..and that special measures to reduce pest prevalence, such as sterile insect releases, are needed to achieve area-wide suppression…” The merits of, or use of, SIT is not a conclusion of this study because no results are given for sterile insect release, as such did not occur. What IS mentioned is the presence of effect of host trees. Stick to the scientific outcomes or conclusions of the particular study, especially in the abstract. The latter is a distillation of the article to be read widely and must be accurate.

Methods

2.2 Traps. It would be useful to know why NZ places traps at 1.5 m height when the international standard is placement as high as possible in the tree. This is particularly the case when detecting low populations of CM, and as has been published from several studies in North America, if not elsewhere.

Results

3.1 “…In non-export orchards, pheromone traps caught an average of 1.62 moths/trap/week, which is 5-fold higher than in the peri-urban traps…” This result (higher in orchards than in peri-urban traps) is most unusual by comparison with other results from other regions, where peri-urban hosts are often untreated and reach high populations. Maybe worthy of more discussion than is provided later on.

Figure 4. What is the point of a large figure showing that traps in host trees caught more moths than traps in non-host trees? It is neither novel nor original. Delete and cite the values in text.

3.2 “…The patchy grassland environment of CHB and trapping densities were different to that of Hastings, as the traps were in a grid in Hastings, whereas in CHB traps were confined to host trees which were had a random distribution…” What is the point of the last phrase in the sentence? Clarify.

Conclusions

“…However, hosts are interspersed with grassland habitat, and eradication could be possible…” As pointed out in the first review, kindly restate conclusions from the present study. This sentence has no meaning because the impact of grassland habitat or mixed landscapes was never examined in the study, and has only been examined in part elsewhere, as in British Columbia. Is politics driving conclusions as in first draft? See below.

Neither the methods and results of the present study nor the conclusions support a key statement in the abstract: that sterile insect release would resolve the problem(s) described in the article. Remove it.

In fact, all of the citations and all prior experience show that sterile insect release in and around peri-urban areas is not in itself enough to drive down such populations. The difficulty of delivering sufficient moths in a timely fashion to eliminate natural CM populations tunneled into the bark of older trees in urban or rural sites, which young adults can mate immediately on emergence from their sites, and are far more competitive than SIT Moths winging through the area, has proven insurmountable to date. Let alone of paying to find and keep access to all of the trees on private properties for long period (see your own survey on access). It is for this reason that the reviewer suggested a discussion of how this was reportedly achieved for other moth(s) in NZ may be instructive.

Funding

It is necessary to list any funds in part or whole from any agency other than internal funding. Did the work occur during periods when at least one author received related funds for consultancies or research from the IAEA or its Insect Control section, whose sole purpose is to promote SIT? Similarly, would some of the data have been removed concerning export orchards (as described) if the export industry were not involved or funding part of it? Please check with authors and correct. See the ethics sections in many bioscience and medical journals for guidance.

Comments, following first review

The comment made by this reviewer to examine carefully the work of Geier is because his is amongst the only work available (apart from some in Canada) regarding the population dynamics of CM at very low levels, and the amazing way in which it can linger at very low levels, and increase immensely, when almost absent. Not because of urban release, which has been attempted only in Canada, and which led to the release devices and sites being derisively named as "bird feeders" by the staff concerned.

Finally, the senior author(s) is/are gently cautioned (as in the prior review) to keep to the science within the study reported. Continue to avoid pandering in any way to demands of politics or industry funding agencies with their agendas.